# Discrimination of Native Vowels in Bilingual and Monolingual Czech Infants with Familial Risk of Dyslexia

Kateřina Kynčlová

Developmental dyslexia is a neurodevelopmental disorder linked to phonological and speech processing deficits, with a strong hereditary component that allows for the allocation of at-risk children with a higher susceptibility to the disorder (Kalashnikova et al., 2020). This study investigates whether Czech at-risk infants show a reduced ability to discriminate native vowels and whether bilingualism modulates the early speech sound development. The study focuses on two types of vowel contrasts that yield different discrimination patterns in typically developing Czech infants–contrasts cued by spectral properties vs. contrasts cued by duration. We hypothesise that at-risk infants will show weaker vowel discrimination, particularly for length contrasts, and that bilingual at-risk infants will exhibit smaller discrimination deficits than monolingual at-risk infants, potentially due to enhanced cognitive and perceptual skills.

The study aims to test 96 Czech infants aged 3.5-7.5M, divided into monolingual (n=48; 24 at-risk) and bilingual (n=48; 24 at-risk) groups. Infants will be tested in a central fixation paradigm. They will be familiarised with strings of naturally produced Czech syllables [fa], and after reaching a habituation criterion, their discrimination of Czech phonemic [fa]-[fe] and [fa]-[fa:] contrasts will be tested with alternating vs. non-alternating trials.

We predict impaired discrimination of vowel length in at-risk infants, consistent with studies showing deficits in consonant length discrimination in at-risk children in languages with phonemic consonantal length (Gerrits et al., 2008; Richardson et al., 2003). Alternatively, at-risk infants acquiring Czech may not be disadvantaged in vowel length discrimination, as this ability seems to be developed in typically developing Czech infants from a very early age (Chládková et al., 2021; Paillereau et al., 2021). Literature on early speech perception development in at-risk bilinguals is lacking and the present results will thus profoundly broaden our understanding of early speech development in bilingual at-risk infants.

**Keywords:** Speech perception, vowel discrimination, developmental dyslexia, at-risk infants, early bilingualism

(This abstract is intended for submission to the *Workshop on Infant Language Development (WILD) 2025*.)

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
