# OpenReview forum: "Discrimination of Native Vowels in Bilingual and Monolingual Czech Infants with Familial Risk of Dyslexia"
_CUNI.cz/2024/CJOLPhD — CUNI 2024 CJOLPhD Submission_

### Official Review · ~Maria_Onoeva1 · 2025-01-06
**Good job!**

This is a nice and clear abstract for the Workshop on Infant Language Development (WILD) 2025. Given the limited space, it’s well-structured and includes all the necessary background and methodological details. I learned a lot about child speech processing and would love to hear more. Best of luck with the submission and the research, I’m looking forward to the results!

I have the following questions to the research:
1) What are the languages for bilingual infants? Czech + another?
2) Are you going to test these syllables only? Why not to test words where these syllables occur in? I guess this ensures clearer results but in real life people do not produce syllables only.
3) Is it possible to track and test these infants further, e.g., when they are todlers? They are at-risk but it doen't mean they will necessary inherit dyslexia, right? Or maybe it works in a different way, I have to learn more.

---

### Official Review · ~Lucie_Jarůšková1 · 2025-01-06
**Great abstract**

The psycholinguistic topic of the paper (*dyslexia and early language perception*) is current, interesting and socially relevant. I especially appreciate that the author does not only examine comparisons between at-risk and typically developing monolingual children, but also includes the bilingual population.

The abstract is written in clear, professional, and excellent English.

In terms of structure, the author clearly defines the issue in the introduction and moves smoothly into a description of the method. If the space of the text allowed and the author had it planned, it would have been useful to give more details about the experiment itself, for example how long it takes, or how many *trials* (items, stimuli) it contains in total –⁠⁠⁠⁠⁠⁠ a few words would have sufficed. Since this is not a finished experiment with results, but a design proposal, this section is essential.
Hypotheses/predictions are listed in two places, they are a bit repetitive, but I understand the repetition in the part of conclusion, as there is an explanation based on literature. It would be appropriate to add to the second hypothesis whether the differences are supposed to be in one of the contrasts (length or spectral quality) or both contrasts –⁠⁠⁠⁠⁠⁠ by adding e.g. *in both contrasts*.

My question:
1. Will the author also compare non-at-risk infants, typically developing monolingual and bilingual?

Overall, this is an excellent and readable abstract that summarizes and presents an interesting experiment, suitable for the conference. Fingers crossed with acceptance!

---

### Official Review · ~Anna_Staňková1 · 2025-01-07
**Great!**

This abstract is well-written and well-structured. I especially like how it is simple to understand even for someone who is not an expert on the topic. Minor comment: I would suggest placing the very last sentence ("Literature on early speech perception ... in biligual at-risk infants") to the first paragraph, maybe in a reformulated way. To me it sounds as an important reason why you are doing this research, so I would stress it by moving it rather to the beginning of the abstract.

---

### Official Review · ~Radek_Šimík1 · 2025-01-08
**Excellent abstract!**

This is an excellent abstract. Each sentence has its clear function in the structure of abstract, in line with the standard structure. The abstract is sufficiently short and clearly structured so that section labeling is not necessary. Depending on whether the results are in place before submission, you might consider distributing some of the contents from the last paragraph to previous sections.